# Improving Mutual Information based Feature Selection by Boosting Unique Relevance

## Abstract

Mutual Information (MI) based feature selection makes use of MI to evaluate each feature and eventually shortlist a relevant feature subset, in order to address issues associated with high-dimensional datasets. Despite the effectiveness of MI in feature selection, we have noticed that many state-of-the-art algorithms disregard the so-called unique relevance (UR) of features, which is a necessary condition for the optimal feature subset. In fact, in our study of seven state-of-the-art and classical MIBFS algorithms, we find that all of them underperform as they ignore UR of features and arrive at a suboptimal selected feature subset which contains a non-negligible number of redundant features. We point out that the heart of the problem is that all these MIBFS algorithms follow the criterion of Maximize Relevance with Minimum Redundancy (MRwMR), which does not explicitly target UR. This motivates us to augment the existing criterion with the objective of boosting unique relevance (BUR), leading to a new criterion called MRwMR-BUR. We conduct extensive experiments with several MIBFS algorithms with and without incorporating UR. The results indicate that the algorithms that boost UR consistently outperform their unboosted counterparts in terms of peak accuracy and number of features required. Furthermore, we propose a classifier based approach to estimate UR that further improves the performance of MRwMR-BUR based algorithms.

## 1 Introduction

High-dimensional datasets tend to contain irrelevant or redundant features, leading to extra computation, larger storage, and decreased performance (Bengio et al., 2013; Gao et al., 2016; Bermingham et al., 2015; Hoque et al., 2016). Mutual Information (MI) (Cover & Thomas, 2006) based feature selection, which is a classifier independent filter method, addresses those issues by selecting a relevant feature subset. We start this paper by discussing the value of MI based feature selection (MIBFS).

**Interpretability**: Dimensionality reduction methods consist of two classes: feature extraction and feature selection. Feature extraction transforms original features into new features with lower dimensionality (e.g., PCA). This method may perform well in dimensionality reduction, but the extraction process (e.g., projection) loses the physical meaning of features (Chandrashekar & Sahin, 2014; Sun & Xu, 2014; Nguyen et al., 2014; Gao et al., 2016). In contrast, feature selection preserves the interpretability by selecting a relevant feature subset. This helps to understand the hidden relationship between variables and makes techniques such as MIBFS preferred in various domains (e.g., healthcare) (Kim et al., 2015; Liu et al., 2018; Chandrashekar & Sahin, 2014).

**Generalization**: Feature selection methods are either classifier dependent or classifier independent (Guyon & Elisseeff, 2003; Chandrashekar & Sahin, 2014). Examples of the former type include the wrapper method and the embedded method (e.g., LASSO (Hastie et al., 2015)) which performs feature selection during the training of a pre-defined classifier. The classifier dependent method tends to provide good performance as it directly makes use of the interaction between features and accuracy. However, the selected features are optimized for the pre-defined classifier and may not perform well for other classifiers. The filter method, which is classifier independent, scores each feature according to its relevance with the label. As a filter method, MIBFS quantifies relevance using MI as MI can capture the dependencies between random variables (e.g., feature and label). Consequently, the feature subset selected by MIBFS is not tied to the bias of the classifier and is relatively easier to generalize (Bengio et al., 2013; L. et al., 2011; Meyer et al., 2008).

**Performance**: Although MIBFS is an old idea dating back to 1992 (Lewis, 1992), it still can provide competitive performance in dimensionality reduction (see several recent survey works (Zebari & et al, 2020; Venkatesh & Anuradha, 2019)). We now provide a new perspective using the Information Bottleneck (Tishby et al., 2000) (IB) to explain the superior performance of MIBFS and suggest why MI is the right metric for feature selection. IB was proposed to search for the solution that achieves the largest possible compression, while retaining the essential information about the target and in (Shwartz-Ziv & Tishby, 2017), IB is used to explain the behavior of neural networks. Specifically, let X be the input data to the neural network, Y be the corresponding label and $\tilde{X}$ be the hidden representation of neural networks. Shwartz-Ziv & Tishby (2017) demonstrate that the learning process in neural networks consists of two phases: (i) empirical error minimization (ERM), where $I(\tilde{X}; Y)$ gradually increases to capture relevant information about the label Y. (ii) representation compression, where $I(\tilde{X}, X)$ decreases and $I(\tilde{X}; Y)$ remains almost unchanged, which may be responsible for the absence of overfitting in neural networks.

We note that the goal of MIBFS is to find the minimal feature subset with maximum MI with respect to the label (Brown et al., 2012). Mathematically, the goal can be written as follows.

$$S^* = \arg\min f(\arg\max_{S \subseteq \Omega} I(S; Y)), \tag{1}$$

where $f(A, B, \cdots) = (|A|, |B|, \cdots)$, $|A|$ represents the number of features in $A$ and $\Omega$ is the set of all features, $S \subseteq \Omega$ is the selected feature subset and $S^*$ is the optimal feature subset. In such a manner, MIBFS naturally converts the representation learning process of neural networks to the process of feature selection (if we consider $S$ as a type of hidden representation $\tilde{X}$) and attempts to obtain an equivalent learning outcome. Specifically, maximizing the $I(S; Y)$ corresponds to the ERM phase and minimizing the size of $S$ corresponds to the representation compression phase. We believe this new perspective sheds light on the superior performance of MIBFS in dimensionality reduction and rationalizes the use of MI for feature selection.

We note that finding the optimal feature subset $S^*$ in (1) through exhaustive search is computationally intractable. Therefore, numerous MIBFS algorithms (Meyer et al., 2008; Yang & Moody, 2000; Nguyen et al., 2014; Bennasar et al., 2015; Peng et al., 2005) are proposed and attempt to select the optimal feature subset following the criterion of Maximize Relevance with Minimum Redundancy (MRwMR) (Peng et al., 2005).

In this paper, we explore a promising feature property, called Unique Relevance (UR), which is the key to select the optimal feature subset in (1). We note that UR has been defined for a long time and it is also known as strong relevance (Kohavi & John, 1997). However, only very few works (Liu et al., 2018; Liu & Motani, 2020) look into it and the use of UR for feature selection remains largely uninvestigated. We fill in this gap and improve the performance of MIBFS by exploring the utility of UR. We describe the flow of the remaining paper together with several contributions as follows.

1. We shortlist seven state-of-the-art (SOTA) and classical MIBFS algorithms and uncover the fact that all of them ignore UR and end up underperforming, namely they select a non-negligible number of redundant features, contradicting the objective of minimal feature subset in (1). In fact, it turns out that the minimal feature subset in (1) must contain all features with UR.

2. We point out that, the heart of the problem is that existing MIBFS algorithms following the criterion of MRwMR (Peng et al., 2005), which lacks a mechanism to explicitly identify the UR of features. This motivates us to augment MRwMR and include the objective of boosting UR, leading to a new criterion for MIBFS, called MRwMR-BUR.

3. We estimate UR using the KSG estimator (Kraskov et al., 2004) and conduct experiments with five representative MIBFS algorithms on six datasets. The results indicate that the algorithms that boost UR consistently outperform their unboosted counterparts when tested with three classifiers.

4. We improve MRwMR-BUR by proposing a classifier based approach to estimate UR and our experimental results indicate that this approach further improves the classification performance of MRwMR-BUR based algorithms.

## 2 BACKGROUND AND DEFINITIONS

We now formally define the notation used in this paper. We denote the set of all features by $\Omega = \{X_k, k = 1, \cdots, M\}$, where $M$ is the number of features. The feature $X_k \in \Omega$ and the label $Y$

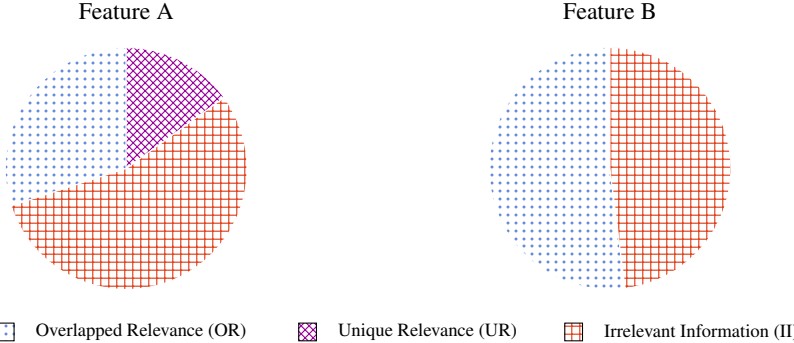

Figure 1: Entropy of feature A and B from the perspective of OR, UR and II. Feature A represents a feature with UR and OR with respect to the label, Feature B represents a feature with no UR and some OR with respect to the label.

are both vectors of length $N$, where $N$ is the number of samples. Let $S \subseteq \Omega$ be the set of selected features and $\tilde{S} \subseteq \Omega$ be the set of unselected features, i.e., $\Omega = \{S, \tilde{S}\}$.

## 2.1 INFORMATION CONTENT: OR, UR, II

The information content presents in a feature can be divided into three parts: Overlapped Relevance, Unique Relevance and Irrelevant Information (see Feature A in Fig. 1).

Relevant information is quantified using MI with the label. The Unique Relevance (UR) of a feature $X_k$ is defined as the unique relevant information which is not shared by any other features in $\Omega$. Mathematically, UR can be calculated as the MI loss when removing that feature from $\Omega$. By the chain rule for MI (Cover & Thomas, 2006), UR can be written as

$$\text{UR} = I(\Omega; Y) - I(\Omega \backslash X_k; Y) = I(X_k; Y | \Omega \backslash X_k). \tag{2}$$

We note that UR is equivalent to strong relevance defined in (Kohavi & John, 1997; Brown et al., 2012). The Overlapped Relevance (OR) of a feature $X_k$ is the relevant information content of a feature $X_k$ which is shared (or overlapped) with other features in $\Omega$. By the chain rule for MI, OR can be written as

$$\text{OR} = I(X_k; Y) - (I(\Omega; Y) - I(\Omega \backslash (X_k); Y)). \tag{3}$$

The definition of OR is equivalent to weak relevance defined in (Kohavi & John, 1997; Brown et al., 2012). Consider an example of $\Omega$ containing two features $X_j$ and $X_k$, then the OR of feature $X_k$ is $I(X_j; X_k; Y)$, which is known as multivariate mutual information. A positive value of $I(X_j; X_k; Y)$ is a sign of redundancy, while a negative value expresses synergy (McGill, 1954).

Irrelevant information (II) can be understood as the noise in the signal. Overfitting to the irrelevant aspects of the data will confuse the classifier, leading to decreased accuracy (John et al., 1994; Song et al., 2011). Mathematically, we define II of feature $X_k$ as

$$\text{II} = H(X_k) - I(X_k; Y) = H(X_k | Y). \tag{4}$$

We note that a feature $X_k$ can be completely irrelevant with respect to the label Y if $I(X_k; Y) = 0$.

There is another popular type of decomposition called partial information decomposition (PID) (Williams & Beer, 2010) which decomposes the total mutual information of a system into three parts: unique information, redundant information, synergistic information and a follow-up work (Bertschinger & et al 2014) attempts to quantify each term based on ideas from decision theory. The definition of UR is the same as the unique information in (Williams & Beer, 2010), but calculated differently from (Bertschinger & et al 2014). Furthermore, OR is equal to the difference of shared information and synergistic information in (Bertschinger & et al 2014).

## 2.2 ESTIMATION OF MUTUAL INFORMATION

In this paper, we estimate MI using the KSG estimator (Kraskov et al., 2004) which uses the $K$ nearest neighbors of points in the dataset to detect structure in the underlying probability distribution.

|             | Gas sensor | Colon | Sonar | Madelon | Leukemia | Isolet |
|-------------|------------|-------|-------|---------|----------|--------|
| Features    | 128        | 2000  | 60    | 500     | 7070     | 617    |
| Instances   | 13874      | 62    | 208   | 2600    | 72       | 1560   |
| Classes     | 6          | 2     | 2     | 2       | 2        | 26     |
| Data Type   | Cont.      | Disc. | Cont. | Cont.   | Disc.    | Cont.  |
| UR (%)      | 2.34%      | 4.3%  | 28.3% | 29.6%   | 34.9%    | 37.1%  |

Table 1: Information of experimental datasets showing the fraction of features with UR.

In a recent work (Gao et al., 2018), the KSG estimator is proven to be consistent under some mild assumptions. We note that the KSG estimator is not applicable when the random variable being studied is a mixture of continuous and discrete values. For the case with mixed random variables, we can apply the mixed KSG estimator (Gao & et al, 2017), which demonstrates good performance at handling mixed variables. We note that the features of all datasets studied in this paper are either purely discrete (real-valued) or continuous while all labels are purely discrete (real-valued) (see Table 1). Hence, we use the KSG estimator (Kraskov et al., 2004) to compute MI quantities.

## 3 A New Criterion for MI based Feature Selection

In this section, we first show the crucial role of UR in selecting the optimal feature subset in (1). Next, we conduct experiments to uncover the fact that all studied MIBFS algorithms are underperforming and hence, motivating MRwMR-BUR as a new criterion for MIBFS.

### 3.1 A Necessary Condition for Optimality

Recall the goal of MIBFS in (1) is to find the minimal feature subset with maximum MI with respect to the label (Brown et al., 2012). Several works (Yu & Liu, 2004; John et al., 1994) have noticed that UR is a necessary condition for the optimal solution in (1). This can be simply proved as follows.

**Proposition 1.** *The optimal feature subset $S^*$ in* (1)*, which has maximum $I(S; Y)$ with minimum $|S|$, must contain all features with UR.*

*Proof.* Assume there exists a feature $X_k \in \Omega$ with non-zero UR. Suppose we have a feature subset $S \subseteq \Omega \backslash X_k$ which has maximum $I(S; Y)$ with minimum $|S|$. Since $X_k$ has non-zero UR, we have $I(\Omega; Y) > I(\Omega \backslash X_k; Y)$ by definition. Therefore, $I(\Omega; Y) > I(S; Y)$ as $I(\Omega \backslash X_k; Y) \geqslant I(S; Y)$ given $S \subseteq \Omega \backslash X_k$. But this contradicts the initial assumption that $I(S; Y)$ is maximum. $\square$

We note that the optimal feature subset $S^*$ may also contain features with OR and no UR at certain situations. For example, consider a feature subset $T$ which contains all features with UR, it is possible to have another feature $X_m \notin T$ which contributes to higher joint MI (i.e., $I(X_m, T; Y) > I(T; Y)$).

### 3.2 Motivating the MRwMR-BUR Criterion

In this subsection, we shortlist seven SOTA and classical MIBFS algorithms: Mutual Information Maximization (MIM) (Lewis, 1992), Joint Mutual Information (JMI) (Yang & Moody, 2000), Joint Mutual Information Maximization (JMIM) (Bennasar et al., 2015), minimal Redundancy Maximum Relevance (mRMR) (Peng et al., 2005), greedy search algorithm (GSA) (Brown et al., 2012). Mutual Information Convex Optimization (MICO) (Sun & et al, 2019), Conditional Mutual Information based Feature Selection considering Interaction (CMIFSI) (Tang et al., 2019) We simulate their feature selection process using Sonar dataset (Dua & Graff, 2017) and evaluate the variation of joint MI $I(S; Y)$ as more features are selected.

In Fig. 2, we classify features as three types: (i) features with nonzero UR, (ii) features with zero UR, and (iii) redundant features, namely that all redundant features (e.g., $X_{redun}$) can be removed after saturation of joint MI without decreasing the joint MI (i.e., $I(S_{satur}; Y) = I(S_{satur} \backslash X_{redun}; Y)$). In Fig. 2 (left), we provide an example of the GSA feature selection process and summarize the performance of all algorithms in Fig. 2 (right), where redundancy rate represents the percentage of redundant features in the selected feature subset when the joint MI saturation is firstly reached.

In Fig. 2 (left), we observe that GSA is able to perform well at the beginning and select features with UR. However, as more features are selected, the joint distribution of the selected feature subset

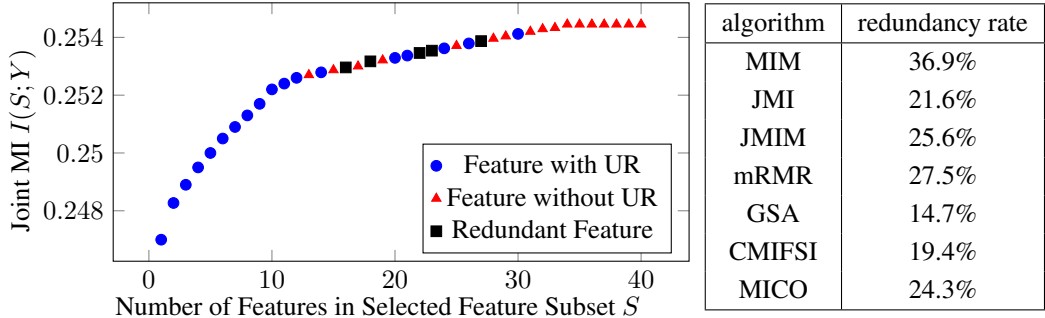

Figure 2: (left) Feature selection with GSA. (right) Redundancy rates for various algorithms.

becomes more complex, and then GSA tends to select some redundant features. Similar performance can be observed for other algorithms (see Fig. 2 (right)). We note that, redundant features contribute nothing, but increase the size of the selected feature subset, undermining the objective of minimal feature subset requirement in (1). Surprisingly, all of the SOTA and classical algorithms studied are selecting a non-negligible number of redundant features, which uncovers the fact that all of them are underperforming. Lastly, we note that similar performance can be observed on another dataset (Gas Sensor dataset (Alexander et al., 2012)) as well (see A.1 in Appendix).

The heart of the problem is that features with UR are not prioritized during the selection process as all these algorithms follow the MRwMR criterion which lacks a mechanism to identify UR. Furthermore, based on our analysis of various datasets (see Table 1), features with UR usually make up a very small fraction of the total number of features and hence, is difficult to select them without explicitly targeting them. This motivates us to augment MRwMR and include the objective of boosting unique relevance (BUR), leading to a new criterion, called MRwMR-BUR. In Section 6, we discuss results showing that the new criterion can significantly reduce the redundancy rate.

## 4 PERFORMANCE EVALUATION

In this section, we shortlist five representative MIBFS algorithms and modify them to include the objective of BUR. We conduct experiments to compare the performance of each algorithm before and after boosting UR, so as to evaluate the proposed MRwMR-BUR criterion.

### 4.1 EXPERIMENT SETUP

To support the proposed MRwMR-BUR criterion, we conduct experiments using six public datasets (Peng et al., 2005; Dua & Graff, 2017; Little et al., 2007; Alexander et al., 2012) described in Table 1 and compare the performance of MRwMR-BUR to MRwMR via Support Vector Machine (SVM) (Cortes & Vapnik, 1995), K-Nearest Neighbors (KNN) (Larose & Larose, 2014) and Random Forest (RF) (Breiman, 2001). We shortlist five representative MRwMR based algorithms: MIM (Lewis, 1992), JMI (Yang & Moody, 2000), JMIM (Bennasar et al., 2015), mRMR (Peng et al., 2005) and GSA(Brown et al., 2012). Those algorithms are augmented to include the objective of BUR as

$$J_{new}(X_i) = (1 - \beta) \times J_{old}(X_i) + \beta \times J_{UR}(X_i), \tag{5}$$

where $J_{old}$ is the MRwMR based algorithm (e.g., MIM) and $J_{new}$ is corresponding MRwMR-BUR form and $J_{UR}(X_i)$ returns the UR of feature $X_i$. We note that this modification is slightly different for JMI as JMI is not bounded and the score increases as more features are selected. Therefore, we divide the original form of JMI by the size of the selected feature subset and include BUR as shown in (5). Moreover, the details for each algorithm with and without BUR are provided in Appendix A.2. We will denote the algorithm that extends XYZ as XYZ_BUR (e.g., MIM and MIM_BUR).

For each run, the dataset is randomly split into two subsets: training dataset (75%) and testing dataset (25%). We apply all MRwMR and MRwMR-BUR (with $\beta = 0.1$) based algorithms on the same training dataset to select features and evaluate them using the same testing dataset. To ensure fair comparison, the parameters of all classifiers are tuned via grid search and we note that all methods share the same grid search range and step size. Some key parameters are tuned as follows. (i) the number of neighbors $K$ for KNN is tuned from 3 to 30 with step size of 2. (ii) the regularization coefficient $c$ for SVM is chosen from $\{0.01, 0.1, 1, 10\}$. (iii) the number of trees in the RF is chosen from $\{10, 15, ..., 50\}$. Lastly, the source code will be released at the camera ready stage.

## 4.2 PERFORMANCE COMPARISON

The results averaged over 20 runs via RF are shown in Fig. 3. For each algorithm, we report two results: (i) the peak averaged accuracy and its spread over 20 runs using boxplot. (ii) the number of features required to obtain the peak accuracy. Moreover, to better summarize the performance difference between the MRwMR based algorithm and its MRwMR-BUR form, we report two sets of results for each dataset: the peak averaged accuracy improvement and the feature improvement. Specifically, as shown below the subplot of Fig. 3, from left to right, it is the minimum, maximum and averaged improvement over five comparisons between the MRwMR based algorithm and its MRwMR-BUR form (e.g., compare MIM to MIM_BUR, JMI to JMI_BUR and so on). A negative value of the improvement indicates that the peak averaged accuracy of the MRwMR-BUR based algorithm is lower than its original form or the number of features required for the MRwMR-BUR based algorithm is more than its original form.

In terms of MRwMR based algorithms, we observe that MIM tends to provide the worst performance in terms of peak averaged accuracy and number of features required (see Figure 3 (a), (b), (c), (f)). We suspect it is because that MIM assumes features are independent from each other, leading to degraded performance. For other MRwMR based algorithms, JMI and GSA generally perform better than other MRwMR based algorithms (see Figure 3 (a), (d), (d), (f)). This finding agrees with (Brown et al., 2012; Liu et al., 2018) as GSA is greedy in nature and JMI can increase the complementary information between features.

Comparing the performance of MRwMR to MRwMR-BUR, we observe that the performance of most MRwMR based algorithms is improved after including the BUR objective (e.g., compare MIM to MIM_BUR). For example, the maximum and average accuracy improvement for the Leukemia dataset is approximately 0.014 and 0.005, respectively. Moreover, fewer number of features (i.e., 9.8 features on average) are required. Similar improvements can be observed for the other datasets as well. We note that similar trends can be found using SVM and KNN as well (see A.3 in Appendix).

## 5 ESTIMATING UR VIA A CLASSIFIER BASED APPROACH

In this section, we first introduce an alternative classifier based approach to estimate UR in Section 5.1. Next, we compare it to performance of estimating UR using the KSG estimator in Section 5.2.

### 5.1 THE CLASSIFIER BASED APPROACH TO ESTIMATE UR

The UR of feature $X_k$ in (2) can be equivalently expressed as

$$I(X_k; Y|\Omega \backslash X_k) = H(Y|\Omega \backslash X_k) - H(Y|\Omega). \tag{6}$$

We note that the term $H(Y|\Omega)$ in (6) is constant for every candidate feature during the feature selection process. Therefore, boosting UR is equivalent to boosting $H(Y|\Omega \backslash X_k)$, which is defined as

$$H(Y|\Omega \backslash X_k) = -\mathbb{E}\{\log P(Y|\Omega \backslash X_k)\}. \tag{7}$$

The RHS of (7) is a function of the likelihood $P(Y|\Omega \backslash X_k)$, which we can estimate using a classifier $\mathbb{Q}$ with parameter $\theta$. Hence, the estimated UR of feature $X_k$ can be written as

$$\text{UR} \equiv H(Y|\Omega \backslash X_k) \approx \frac{1}{N} \sum_{i=1}^{N} \mathbb{Q}(y^i|(\Omega \backslash x_k)^i, \theta), \tag{8}$$

where $N$ is the number of training samples. The reason for estimating UR via a classifier is two-fold: (i) We understand that the estimation of high-dimensional MI is still in the development stage. This approach provides an alternate way to estimate UR and the estimated UR will approach the true value as the number of samples $N$ grows given the classifier $\mathbb{Q}$ is a consistent estimator (Lehmann & Casella, 2006). (ii) We note that different classifiers may favour the UR of features differently due to their working mechanisms and assumptions. For example, the improvement of MRwMR-BUR over MRwMR is marginal using SVM with linear kernel (see A.3 in Appendix). We suspect that this is because MI quantifies a non-linear relationship between random variables and this non-linear relationship may not be of much help to the linear classifier. Estimating UR via a classifier attempts to adapt the value of UR to different classifiers. Furthermore, we note that estimating UR in such a manner changes MRwMR-BUR based algorithms to a classifier dependent method, but we highlight that the selected feature subset still preserves the interpretability and is preferred in some performance-oriented tasks if the classifier based approach can further improve the performance.

|  | Gas sensor | Colon | Sonar | Madelon | Leukemia | Isolet |
|---|---|---|---|---|---|---|
| MIM_BUR | **99.45%**(88) | 85.21%(23) | 80.23%(41) | 72.75%(77) | 96.64%(42) | 86.34%(139) |
| MIM_BUR_CLF | 99.32%(68) | **86.41%**(19) | **80.57%**(47) | **74.32%**(57) | **98.64%**(38) | **87.69%**(131) |
| mRMR_BUR | 99.48%(71) | 87.09%(7) | 81.29%(40) | 72.57%(77) | 96.63%(63) | 86.13%(131) |
| mRMR_BUR_CLF | **99.50%**(69) | **88.82%**(31) | **81.53%**(37) | **73.19%**(57) | **99.13%**(53) | **88.50%**(135) |
| JMI_BUR | 99.44%(75) | 74.93%(39) | 81.25%(48) | 71.93%(63) | 97.24%(75) | **87.94%**(143) |
| JMI_BUR_CLF | **99.48%**(72) | **75.76%**(36) | **81.79%**(40) | **72.47%**(57) | **99.33%**(64) | 87.58%(127) |
| JMIM_BUR | **99.48%**(80) | 75.92%(19) | 80.72%(34) | **73.32%**(67) | 95.88%(61) | 85.44%(123) |
| JMIM_BUR_CLF | 99.45%(79) | **76.31%**(16) | **81.13%**(46) | 73.19%(57) | **99.42%**(72) | **85.72%**(127) |
| GSA_BUR | 99.47%(87) | 86.49%(13) | **81.94%**(26) | 72.93%(59) | 95.74%(58) | 87.25%(121) |
| GSA_BUR_CLF | **99.50%**(83) | **88.25%**(21) | 81.73%(49) | **74.58%**(57) | **99.21%**(65) | **88.97%**(106) |

Table 2: Peak averaged accuracy and corresponding number of features required for MRwMR-BUR based algorithms (with $\beta = 0.1$) using Random Forest. The algorithm with an extension of CLF indicates that the algorithm uses the classifier based approach to estimate UR.

## 5.2 PERFORMANCE COMPARISON

We apply this new approach to estimate UR and re-implement the MRwMR-BUR based algorithms in Section 4. We highlight that all estimated URs are first normalized from 0 to 1 using min-max normalization. The peak averaged accuracy and corresponding number of features required are shown in Table 2 with $\beta = 0.1$. We note that the UR is estimated using the classifier being tested. In Table 2, the KSG estimator is used to estimate UR except for the CLF named algorithms, which use the classifier based approach. We highlight the best accuracy using shading. In general, we find that the classifier based approach outperforms the KSG estimator in terms of peak averaged accuracy and number of features required in most datasets and algorithms studied. For example, the accuracy of the classifier based approach for mRMR-BUR on the Leukemia dataset is increased by 0.025 while 10 fewer features are required. Furthermore, similar performance trends can be observed using KNN and SVM as well (see A.3 in Appendix).

## 6 REFLECTIONS

In this section, we present some reflections and suggestions for future work.

**Potentially Better MIBFS Algorithms:** We highlight that, in this paper, we are not proposing a new MIBFS algorithm. Instead, we explore a new criterion for the design of MIBFS algorithms that incorporate UR into the objective. Our experimental results demonstrate that the proposed MRwMR-BUR criterion has superior performance over the existing MRwMR criterion. We believe this new insight can inspire better MIBFS algorithms without increasing algorithm complexity. In fact, the UR only needs to be calculated once and can be reused in subsequent calculations. Hence, the complexity of the UR augmented algorithms is comparable to the original algorithms.

**Redundancy Rates:** By not prioritizing UR, existing MIBFS algorithms choose a high fraction of redundant features and end up underperforming. Our experiments in Section 3.2 demonstrate that the redundancy rate of GSA is 14.7%. By incorporating UR, the redundancy rate of GSA_BUR is significantly reduced to 9.09% with $\beta = 0.05$ and 6.25% with $\beta = 0.1$.

**Always Select Features with UR First?**: The results in Section 3.1 show that the minimal feature subset $S^*$ in (1) must include all features with UR. However, it may not be a good idea to include only features with UR at first when the selected feature size is constrained. This is because features with UR (e.g., $X_k$) may not contribute to higher relevance in the short term given $S \subseteq \Omega \backslash X_k$ (e.g., $I(X_k; Y|S) = 0$) as removing conditional features may decrease the MI.

**Optimal Value of $\beta$:** The performance of MRwMR-BUR based algorithms relies on the value of $\beta$, which balances the original objective of maximizing relevance with prioritizing UR. From extensive experiments, we have found that using $\beta = 0.1$ to balance UR and relevance. Alternatively, $\beta$ can be thought of as a hyper-parameter and tuned via a validation dataset. The theoretical determination of the optimal value $\beta$ is clearly worth deep thought. This could help to further minimize redundancy by prioritizing more important features.

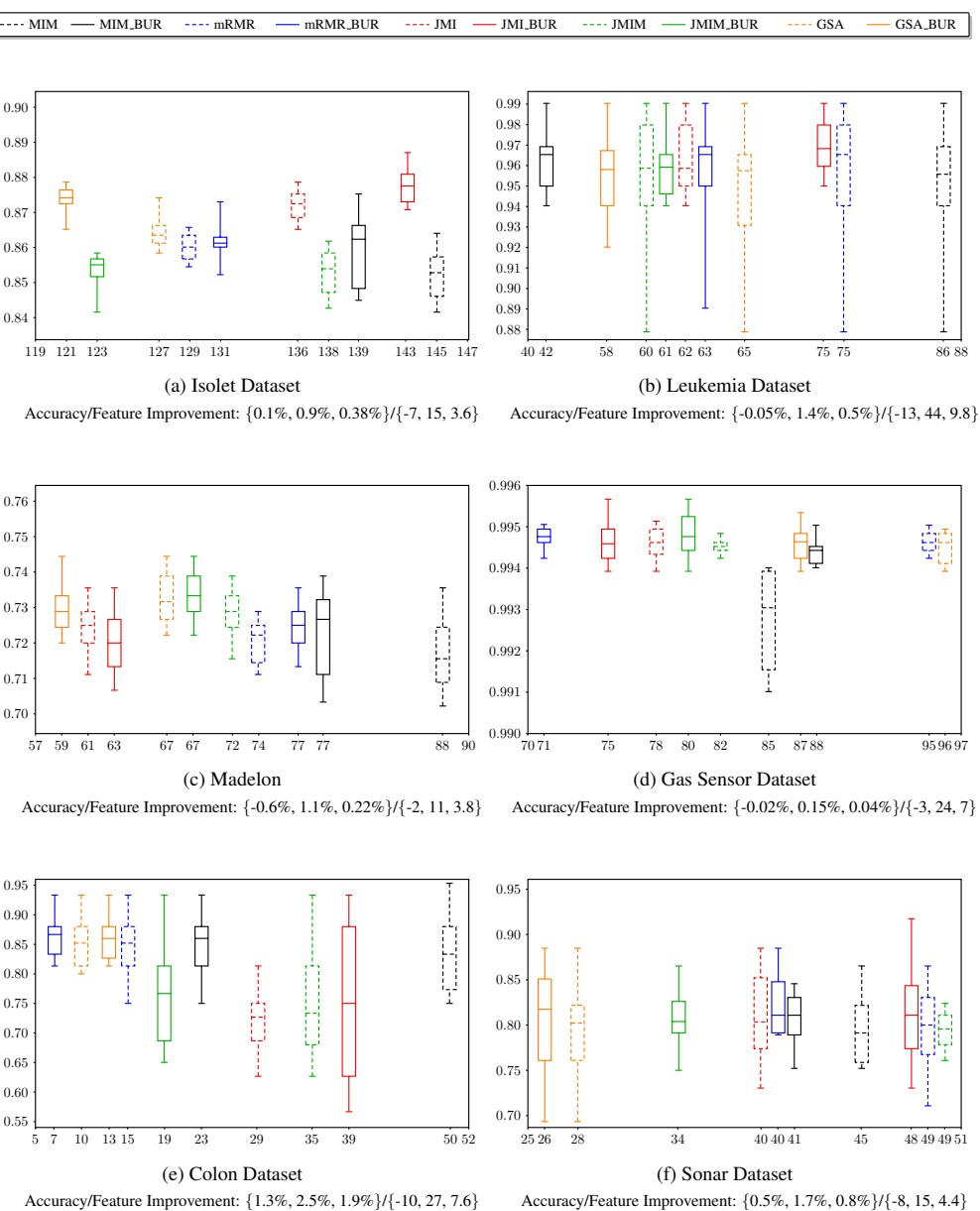

Figure 3: Performance Comparison between MRwMR based algorithms and corresponding MRwMR-BUR forms on six public datasets. The vertical and horizontal axes are classification accuracy and the corresponding number of features required, respectively. The results shown are averages over 20 trials via Random Forest with $\beta = 0.1$. The box corresponds to the lower and upper quartile. The horizontal line inside the box depicts the peak averaged accuracy and the whiskers depict the minimum and maximum over the trials. In each subplot, we report two sets of results: peak averaged accuracy improvement and feature improvement. Specifically, from left to right, it is the minimum, maximum and averaged improvement over five comparisons between the MRwMR based algorithm and its MRwMR-BUR form (e.g., compare MIM to MIM_BUR, JMI to JMI_BUR and so on). A negative value of improvement indicates that the peak averaged accuracy of the MRwMR-BUR algorithm is lower than its original form or the number of features required for the MRwMR-BUR algorithm is more than its original form. Similar Performance can be observed for SVM and KNN as well (see A.3 in Appendix).

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

# A APPENDIX

In A.1, we provide more results of the redundancy experiment conducted in Section 3.2. Next, we show the scoring function of studied MIBFS algorithms before and after including BUR in A.2. Lastly, we show the performance comparison between MRwMR and MRwMR-BUR, as well as the effect of the classifier based approach using SVM and KNN in A.3.

## A.1 REDUNDANCY EXPERIMENT ON THE GAS SENSOR DATASET

In Section 3.2, we conduct experiments on the Sonar dataset to demonstrate that all studied MIBFS algorithms are selecting redundant features, undermining the objective of minimal feature subset in (1). In this subsection, we conduct the same experiment using the Gas Sensor dataset and summarize the performance in Table 3.

| algorithm | redundancy rate |
|-----------|-----------------|
| MIM | 23.6% |
| JMI | 22.5% |
| JMIM | 24.44% |
| mRMR | 18.6% |
| GSA | 15.4% |
| CMIFSI | 27.7% |
| MICO | 20.5% |

Table 3: Redundancy Experiment Results on the Gas Sensor Dataset

## A.2 SCORING FUNCTIONS OF THE FIVE MRwMR ALGORITHMS WITH/WITHOUT BUR

In this subsection, we follow the same notation as in Section 2. Namely, $X_i$, $Y$, $S$, and $\tilde{S}$ are the feature, the label, the set of selected features, and the set of unselected features, respectively. $J_{\text{UR}}(X_i)$ denotes the unique relevance of feature $X_i$ (i.e., $J_{\text{UR}}(X_i) = I(\Omega; Y) - I(\Omega \backslash (X_i); Y)$). The five representative MRwMR algorithms are: MIM Lewis (1992), JMI Yang & Moody (2000), JMIM Bennasar et al. (2015), mRMR Peng et al. (2005) and GSABrown et al. (2012). The scoring functions for these algorithms with and without BUR are shown below.

$$J_{\text{MIM}}(X_i) = \arg\max_{X_i \in \tilde{S}} \ I(X_i; Y) \tag{9}$$

$$J_{\text{MIM\_BUR}}(X_i) = \arg\max_{X_i \in \tilde{S}} \ (1-\beta) \times I(X_i; Y) + \beta \times J_{\text{UR}}(X_i) \tag{10}$$

$$J_{\text{JMI}}(X_i) = \arg\max_{X_i \in \tilde{S}} \ \sum_{X_j \in S} I(X_i, X_j; Y) \tag{11}$$

$$J_{\text{JMI\_BUR}}(X_i) = \arg\max_{X_i \in \tilde{S}} \ (1-\beta) \times \sum_{X_j \in S} I(X_i, X_j; Y) \times \frac{1}{|S|} + \beta \times J_{\text{UR}}(X_i) \tag{12}$$

$$J_{\text{mRMR}}(X_i) = \arg\max_{X_i \in \tilde{S}} \ I(X_i; Y) - \frac{1}{|S|} \sum_{X_j \in S} I(X_i, X_j) \tag{13}$$

$$J_{\text{mRMR\_BUR}}(X_i) = \arg\max_{X_i \in \tilde{S}} \ (1-\beta) \times \left( I(X_i; Y) - \frac{1}{|S|} \sum_{X_j \in S} I(X_i, X_j) \right) + \beta \times J_{\text{UR}}(X_i) \tag{14}$$

$$J_{\text{JMIM}}(X_i) = \arg\max_{X_i \in \tilde{S}} \ \min_{X_j \in S} I(X_i, X_j; Y) \tag{15}$$

$$J_{\text{JMIM\_BUR}}(X_i) = \arg\max_{X_i \in \tilde{S}} \ (1-\beta) \times ( \min_{X_j \in S} I(X_i, X_j; Y)) + \beta \times J_{\text{UR}}(X_i) \tag{16}$$

$$J_{\text{GSA}}(X_i) = \arg\max_{X_i \in \tilde{S}} \ I(X_i, S; Y) \tag{17}$$

$$J_{\text{GSA\_BUR}}(X_i) = \arg\max_{X_i \in \tilde{S}} \ (1-\beta) \times I(X_i, S; Y) + \beta \times J_{\text{UR}}(X_i) \tag{18}$$

## A.3 MORE EXPERIMENTAL RESULTS USING SVM AND KNN

In the main paper, we only provide experimental results using random forest. We note that similar performance trends can be observed using SVM and KNN as well (see SVM in Table 4 and KNN in Table 5).

| | Gas sensor | Colon | Sonar | Madelon | Leukemia | Isolet |
|---|---|---|---|---|---|---|
| MIM | 96.76%(93) | 81.13%(67) | 73.02%(37) | 61.52%(36) | 96.65%(99) | 88.22%(148) |
| MIM_BUR | 96.76%(93) | 82.23%(63) | 73.75%(39) | 61.83%(35) | 97.72%(97) | 88.73%(121) |
| MIM_BUR_CLF | **97.23%**(89) | **82.58%**(65) | **74.28%**(48) | **65.21%**(42) | **98.23%**(81) | **89.48%**(109) |
| mRMR | 96.76%(99) | 83.11%(40) | 75.27%(49) | 62.07%(39) | 97.72%(77) | 89.33%(135) |
| mRMR_BUR | **96.84%**(95) | 82.97%(42) | **75.49%**(43) | 62.07%(39) | 97.72%(77) | **89.62%**(131) |
| mRMR_BUR_CLF | 96.75%(89) | **84.31%**(37) | 75.23%(48) | **62.21%**(47) | **98.63%**(61) | 89.45%(127) |
| JMI | 96.84%(89) | 73.32%(17) | 74.13%(41) | 61.25%(31) | 96.78%(86) | 89.91%(134) |
| JMI_BUR | 96.84%(89) | **73.98%**(15) | 74.25%(37) | **62.09%**(35) | 96.82%(86) | 89.47%(144) |
| JMI_BUR_CLF | **97.23%**(93) | 73.62%(19) | **75.32%**(49) | 61.88%(37) | **98.15%**(75) | **90.08%**(119) |
| JMIM | 96.70%(98) | 76.85%(19) | 73.35%(49) | 62.25%(41) | 96.82%(93) | 88.58%(138) |
| JMIM_BUR | 96.70%(98) | 77.58%(23) | 73.37%(38) | 62.31%(44) | 97.28%(94) | 88.62%(144) |
| JMIM_BUR_CLF | **97.35%**(95) | **79.62%**(27) | **73.62%**(37) | **62.47%**(43) | **98.42%**(71) | **89.21%**(133) |
| GSA | 96.48%(95) | 73.77%(35) | 73.82%(36) | 63.51%(39) | 96.82%(67) | 89.37%(138) |
| GSA_BUR | 96.48%(95) | 74.28%(27) | **74.13%**(31) | 63.27%(33) | 97.14%(54) | 89.75%(131) |
| GSA_BUR_CLF | **96.86%**(96) | **76.33%**(32) | 73.95%(46) | **63.72%**(37) | **98.62%**(68) | **90.02%**(137) |

Table 4: Peak averaged accuracy and corresponding number of features required for each MRwMR and MRwMR-BUR based algorithm ($\beta = 0.1$) using SVM. For MRwMR-BUR based algorithms, the algorithm with an extension of CLF indicates that the algorithm uses the classifier based method to estimate UR. Otherwise, the algorithm estimates UR using the KSG estimator.

| | Gas sensor | Colon | Sonar | Madelon | Leukemia | Isolet |
|---|---|---|---|---|---|---|
| MIM | 99.10%(87) | 84.12%(63) | 82.95%(24) | 73.38%(57) | 95.51%(46) | 78.48%(139) |
| MIM_BUR | **99.15%**(85) | 85.32%(41) | 83.44%(23) | 74.76%(51) | 96.13%(40) | 78.85%(135) |
| MIM_BUR_CLF | 99.08%(80) | **86.42%**(45) | **84.82%**(45) | **77.21%**(67) | **97.82%**(29) | **80.02%**(131) |
| mRMR | 99.06%(83) | 82.57%(45) | 85.02%(30) | 75.53%(61) | 97.72%(54) | 81.02%(129) |
| mRMR_BUR | 99.08%(81) | 85.02%(32) | **85.81%**(31) | 76.21%(67) | 97.70%(53) | 82.13%(117) |
| mRMR_BUR_CLF | **99.21%**(81) | **86.52%**(35) | 84.97%(30) | **76.58%**(69) | **98.61%**(48) | **82.75%**(121) |
| JMI | 99.16%(72) | 70.02%(47) | 83.42%(48) | 79.15%(50) | 95.58%(46) | 79.75%(135) |
| JMI_BUR | 99.14%(65) | 70.58%(43) | **85.02%**(45) | **80.02%**(53) | 96.52%(45) | 79.48%(137) |
| JMI_BUR_CLF | **99.32%**(80) | **71.35%**(41) | 84.39%(48) | 79.72%(59) | **99.13%**(39) | **81.05%**(125) |
| JMIM | **99.12%**(85) | 72.17%(85) | 84.17%(49) | 77.23%(60) | 95.81%(34) | 79.16%(137) |
| JMIM_BUR | 99.09%(80) | 76.22%(63) | 83.82%(47) | 78.36%(63) | 95.83%(34) | 79.48%(139) |
| JMIM_BUR_CLF | 98.97%(81) | **77.58%**(53) | **84.23%**(41) | **79.51%**(53) | **99.47%**(29) | **80.27%**(131) |
| GSA | 98.98%(58) | 85.02%(72) | 83.46%(48) | 78.15%(53) | 95.61%(59) | 79.97%(133) |
| GSA_BUR | 99.01%(56) | 86.22%(61) | 84.62%(40) | 78.42%(55) | 95.84%(62) | 80.25%(121) |
| GSA_BUR_CLF | **99.27%**(67) | **88.53%**(52) | **85.51%**(45) | **78.57%**(49) | **98.63%**(53) | **81.23%**(117) |

Table 5: Peak averaged accuracy and corresponding number of features required for each MRwMR and MRwMR-BUR based algorithm ($\beta = 0.1$) using KNN. For MRwMR-BUR based algorithms, the algorithm with an extension of CLF indicates that the algorithm uses the classifier based method to estimate UR. Otherwise, the algorithm estimates UR using the KSG estimator.

