# OpenReview forum: "Improving Mutual Information based Feature Selection by Boosting Unique Relevance"
_ICLR.cc/2021/Conference — Reject_

### Official Review · AnonReviewer4 · 2020-10-27
**The paper presents an investigation of Mutual information based feature selection methods and the use of unique relevance (UR) with mutual information.**

**Rating:** 4
**Confidence:** 5

**Review:**

The paper presents an investigation of Mutual information based feature selection methods and the use of unique relevance (UR) with mutual information.

**Cons:**
- This paper investigates the feature selection methods based on mutual information and integrates the UR term to the methods.
- The authors compared their method with different Mutual information features selection based methods. The results indicate that the method improves the baseline.
- As mentioned by the authors, using the UR measure can be beneficial in many domains, such expandability of a neural network model.
The experimental section is acceptable.

**Weakness:**
- This paper has substantial overlap with the literature [1-3], especially [1].
- It would be beneficial to mention how this work is different from these papers.
- A Comparison with these methods is missing.

**Minor comments:**
1) In the abstract, mention that MIBFS stands for mutual information based feature selection.
2) Equation (1) is not consistent ( $arg \  min\  f(...)$ )
3) In equation (4), what is function $H()$. A description right after using the function for the first time is beneficial.
4) Selected feature subset -> set of selected features / selected features

**Score:**
- I vote to reject this paper. My main concern is the substantial overlap with the published literature. In my opinion, there is not enough novel material in this paper for this conference.

**Questions:**
1) What is the difference between the method presented in this paper and [1]
2) Provide a comparison with the literature [1-3].

**Additional Feedback:**
- An optimal set of features in the classification does not necessarily contain features with unique relevance to the class labels. For instance, assume two features that do not have any unique information, but their combination provides unique information. The investigation of this issue by using a synthetic data set would be interesting.

**References:**
- [1] Liu, Shiyu, and Mehul Motani. "Exploring Unique Relevance for Mutual Information based Feature Selection." 2020 IEEE International Symposium on Information Theory (ISIT). IEEE, 2020.
- [2] Liu, Shiyu, and Mehul Motani. "Feature selection based on unique relevant information for health data." arXiv preprint arXiv:1812.00415 (2018).
- [3] Liu, Shiyu, et al. "Suri: Feature selection based on unique relevant information for health data." 2018 IEEE International Conference on Bioinformatics and Biomedicine (BIBM). IEEE, 2018.

---

> ### Author Response · Authors · 2020-11-21
> **Response to Reviewer 4**
>
> We would like to thank you for your valuable time and useful suggestions. Please see our responses below.
>
> **1. On the overlap with [Liu, 2020]:**
>
> We do see your concern about the overlap with [Liu, 2018] and [Liu, 2020].  We would like to explain that we viewed the work in [Liu, 2020] as preliminary work for a very different audience and research community. Our current work is a follow-up work which addresses several important limitations of the analysis and the approach in [Liu, 2020] and makes several contributions not in [Liu, 2020]. Please see our general response regarding the difference from [Liu, 2020] and [Liu, 2018]. We hope our explanation suffices and we will endeavor to make the current paper as impactful as possible.
>
> **2. On the Exploration of the Information Synergy:**
>
> Your comment in the additional feedback is very insightful. We agree that information synergy (i.e., the idea that combined features provide more information than the single features provide separately) is an interesting topic to explore, but we note that this does not dilute the importance of UR. The optimal feature subset in (1) must contain all features with UR. Furthermore, we note that the optimal feature subset may contain other features as well. The potential cause could be the information synergy among features.
>
> We would like to thank you again for your valuable time. We apologize for the overlap with previous results. In the revised paper, we will remove the overlapping content, highlight the differences, and address all minor comments. We hope all doubts are now cleared and we are glad to clarify if you have any further questions.

---

### Official Review · AnonReviewer3 · 2020-10-28
**Perspectives and methods are not novel, there is a technical flaw.**

**Rating:** 4
**Confidence:** 5

**Review:**

This work suggests improving mutual informaton based feature selection methods with an extra term (i.e., the unique relevance (UR)), and introduces a hyper-parameter $\beta$ to weight the UR. The work is easy to follow. However, the perspectives and methods are not novel. And there is a technical flaw in the analysis.

1. It seems to me, in terms of methodology, that the difference of this work to [Liu, 2018] and [Liu, 2020] is that this work has two ways to estimate UR, one is based on the KSG estimator, another is based on a classifier. However, the objective (i.e., introducing a weighted term on UR) is not new and is shown in both [Liu, 2018] and [Liu, 2020].

2. Author wants to justify the theoretical guarantee of mutual information based feature selection. However, the perspectives are not new. For example, it is widely acknowledged that feature selection can be interpreted with information bottleneck. There is also a very early work that explicitly implement this idea [1]. On the other hand, author wants to link the feature selection with the state-of-the-art on the learning dynamics of deep neural networks (i.e., the fitting phase and the compression phase). However, the connection seems strange. Note that, one is on the objective itself, another is on the training (or optimization) of the objective.
[1] Hecht, Ron M., and Naftali Tishby. "Extraction of relevant speech features using the information bottleneck method." In Ninth European Conference on Speech Communication and Technology. 2005.

3. The UR terminology is not new, and has been mentioned in very early works in mutual information based feature selection.

4. A technical flaw: I disagree that author mentions that "the UR is the same to the unique information in the partial information decomposition (PID) frameowork". Note that, in PID, there are only three equations but with four unknowns, which makes the estimation of unique information an underdetermined problem (unless we made extra assumptions). However, UR can be simply estimated by its analytical expression.

---

> ### Author Response · Authors · 2020-11-21
> **Response to Reviewer 3 (No technical flaw in the paper’s results)**
>
> We would like to thank you again for your valuable time and insightful comments. Please see our responses below.
>
> **1. On the overlap with [Liu, 2020]:**
>
> We do see your concern about the overlap with [Liu, 2018] and [Liu, 2020].  We would like to explain that we viewed the work in [Liu, 2020] as preliminary work for a very different audience and research community. Our current work is a follow-up work which addresses several important limitations of the analysis and the approach in [Liu, 2020] and makes several contributions not in [Liu, 2020]. Please see our general response regarding the difference from [Liu, 2020] and [Liu, 2018]. We hope our explanation suffices and we will endeavor to make the current paper as impactful as possible.
>
> **2. There is no technical flaw:**
>
> We do not think that there is a technical flaw in the paper’s results. What the reviewer is referring to is a remark comparing UR to the notion of unique information in a previous work. This comparison does not affect the validity of the technical results in the current paper.
>
> We now address the comparison of UR to unique information from [Williams & Beer 2010]. Our definition of UR is the unique information which is not shared by any other features. In the PID paper (Williams & Beer, 2010), they used an example of I(S; R1, R2) to define unique information as the information R1 provides about S that R2 does not, or vice versa. In this sense, the two definitions are the same. We do note that there may be multiple methods to estimate UR, and perhaps, this is what the reviewer is referring to. We do thank you for your thoughtful comment and will reconsider the comparison to unique information carefully.
>
> **3. Information Bottleneck and Feature Selection:**
>
> Thank you for the interesting reference by [Hecht & Tishby 2005], which presents a method to use IB to do feature extraction. We note our work is not about using IB to do feature extraction. Rather we connect the goal of MIBFS (i.e., minimal feature subset with maximum MI) to the learning dynamics of neural networks (i.e., fitting and compression). The former (i.e., MIBFS) attempts to approach the objective by selecting, while the later (i.e., neural networks) attempts to approach the objective by training. To the best of our knowledge, this perspective is not widely known and we believe it may increase understanding of the problem overall.
>
> **4. On the Use of UR Terminology:**
>
> To the best of our knowledge, the very early works on MIBFS [Lewis 1992, Kovahi & John 1994] do not use the terminology unique relevance. Instead [Kohavi & John 1994] use the term strong relevance (we point this out in Section 2.1). These and other previous works do not use UR (or strong relevance) to prioritize features during selection (which is included in the MRwMR-BUR criterion). We would like to emphasize that the contributions of our paper are with respect to: (i)  the important finding that all studied MRwMR based algorithms are underperforming, (ii) the motivating finding that prioritizing UR helps to achieve the objective of MIBFS, (iii) and the proposed classifier based approach to estimate UR.
>
> We would like to thank you again for your valuable time. We apologize for the overlap with previous results. In the revised paper, we will remove the overlapping content, highlight the differences, and address all minor comments. We hope all doubts are now cleared and we are glad to clarify if you have any further questions.

---

### Official Review · AnonReviewer1 · 2020-10-29
**A meaningful job for MI based feature selection**

**Rating:** 8
**Confidence:** 3

**Review:**

In this paper, the authors recognized the function of unique relevance (UR) of features for optimal feature selection and augmented the existing mutual information based feature selection (MIBFS) methods by boosting unique relevance (BUR). As a result, they proposed a new criterion called MRwMR-BUR. Experimental results are provided to show that MIBFS with UR consistently outperform their unboosted conterparts in terms of peak accuracy and number of features required.

Overall,  this paper has thrown new light upon MI based feature selection and the results are valuable. However, I think the paper could be improved from the following two aspects.

1. To their credit, the authors have introduced the background of MI, OR, UR, and II.  However, some of the points are not made clear. For example, at the end of Sec. 3.1, they stated that "We note that the optimal feature subset S* may also contain features with OR and no UR at certain situations. For example.....",   which seems contradictory to the Proposition 1.

2. The authors are suggested to  improve the organization and the presentation of the paper. The current version is not easy to follow. For example, there appears the term J_{UR}(X_i) in Eq. (5), but I do not see any explicit definition of it until I reading Appendix A.2.

---

> ### Author Response · Authors · 2020-11-21
> **Response to Reviewer 1**
>
> We would like to thank you for recognizing our work and useful suggestions.
>
> **Regarding Proposition 1:** We note that the Proposition 1 proves that the optimal feature subset S* must contain all features with UR, but it does not contradict with the fact that S* may contain some other features. In other words, the set containing all features with UR is a necessary subset of the S* and S* may also contain other features.
>
> We hope all doubts are cleared and we will improve the organization and the presentation in the revised paper. We are glad to clarify if you have any further questions.

---

### Official Review · AnonReviewer2 · 2020-10-29
**This paper is a slightly expanded version of work published elsewhere.**

**Rating:** 2
**Confidence:** 5

**Review:**

Update: The author response has not changed my opinion that there is insufficient new material in this paper vs the ISIT paper, and the presentation of the material from the ISIT paper does not note that this material was previously presented there. Without clarity in what is the novel material claimed in this paper it should not be accepted.

This paper presents a modification to existing information theoretic feature selection algorithms which adds a strong relevance term estimated using a k-nn MI estimator. It's a slightly expanded copy of a paper published at IEEE International Symposium on Information Theory 2020, referenced as "Exploring unique relevance for mutual information based feature selection" Liu & Motani 2020. This paper contains the same experimental results, same plots, same theoretical description and is in most ways a direct copy of the ISIT paper, violating ICLR's dual submission policy. I think the only new material is the experimental results on the CLF algorithms.

The authors should note that GSA-BUR has already been published as the SURI technique (referenced as Liu et al 2018), even considering this paper is a version of the ISIT 2020 paper.

The notion of "redundancy rate" is ill-defined, and the experiments which measure it are not discussed. If it's measuring the joint mutual information then it's a measure of the approximation used, and also a factor of the greedy search algorithm (which is used by all the criteria considered in the paper).

The proof of proposition 1 follows from the definition and has been known since the 1990s when strong relevance was introduced.

Given the CLF variants use a classifier to estimate the probabilities then the authors should validate that the features are still widely useful (by transfering the features found using the SVM to the RF) or compare performance against a wrapper like RFE as it's similarly expensive.

---

> ### Author Response · Authors · 2020-11-21
> **Response to Reviewer 2 (Part 2/2)**
>
> **3. On the Redundancy Rate Experiments:**
>
> Redundancy rate is defined on page 4, Sec 3.2: Redundancy rate is the percentage of redundant features included in the selected feature subset S when S first reaches the joint MI saturation (i.e., the first time S obtains the highest joint MI).
>
> The experiments for measuring redundancy rate are defined on page 4, Sec 3.2. We believe the details needed to reproduce the results in Fig. 2 are provided in the paper. The redundancy rate experiment is to investigate the feature subsets selected by the seven feature selection algorithms using the Sonar dataset. Specifically, we calculate the redundancy rate of the feature subset selected by  each algorithm and evaluate their performance in achieving the objective of MIBFS.
>
> **4. On the Novelty of Proposition 1:**
>
> We agree that Proposition 1 is known in the literature and have actually provided relevant citations. In the revised paper, we will include Prop. 1  for completeness but will omit the proof. We would like to emphasize that the contributions of our paper are with respect to: (i)  the important finding that all studied MRwMR based algorithms are underperforming, (ii) the motivating finding that prioritizing UR helps to achieve the objective of MIBFS, (iii) and the proposed classifier based approach to estimate UR.
>
> We would like to thank you again for your critical review. We apologize for the overlap with previous results. In the revised paper, we will remove the overlapping content and highlight the differences. We will update our paper with new results and we are glad to clarify if you have any further questions.

---

> ### Author Response · Authors · 2020-11-21
> **Response to Reviewer 2 (Part 1/2)**
>
> We would like to thank you for your valuable time, insightful feedback and useful suggestions. Please see our responses below.
>
> **1. On the overlap with [Liu, 2020]:**
>
> We do see your concern about the overlap with [Liu, 2018] and [Liu, 2020].  We would like to explain that we viewed the work in [Liu, 2020] as preliminary work for a very different audience and research community. Our current work is a follow-up work which addresses several important limitations of the analysis and the approach in [Liu, 2020] and makes several contributions not in [Liu, 2020]. Please see our general response regarding the difference from [Liu, 2020] and [Liu, 2018]. We hope our explanation suffices and we will endeavor to make the current paper as impactful as possible.
>
> **2. New Experimental Results with RFE:**
>
> We note that the CLF variant changes MIBFS to a semi-wrapper method. Therefore, it is important to compare its performance to other wrapper methods. As suggested, we expanded our comparison to recursive feature elimination (RFE). Specifically, we conduct experiments using the same datasets and same configurations as Section 4.1. We apply RFE with a 10-fold cross validation. Then we shortlist the best feature subset, retrain the model and calculate the test accuracy.  The test accuracy averaged over 5 runs using SVM is shown below.
>
> From the table below, we observe that the performance of RFECV is generally comparable to the MRwMR based algorithms, but underperforms the MRwMR-BUR based algorithms with and without using the CLF variant. We posit this is because the selected features of RFECV are simply identified by their weights and do not take into account any correlation the features may have. This may make it hard for RFECV to generalize well on the unseen data.
>
> =====================================================================================\
> SVM (linear)----------Gas_sensor--------Colon--------Sonar----------Madelon-------Leukemia-------Isolet------\
> MIM--------------------96.7%(93)-------81.1%(67)-----73.0%(37)----61.5%(36)-----96.7%(99)------88.2%(148)\
> MIM_BUR-------------97.1%(83)-------82.2%(63)-----73.8%(39)----61.8%(35)-----97.7%(97)------88.7%(121)\
> MIM_BUR_CLF-------97.2%(89)-------82.6%(65)-----74.2%(48)----65.2%(42)-----98.2%(81)------89.5%(109)\
> -----------------------------------------------------------------------------------------------------------------------------------------\
> mRMR-----------------96.7%(93)-------83.1%(40)-----75.2%(49)-----62.0%(39)-----97.7%(77)------88.3%(135)\
> mRMR_BUR----------96.8%(83)-------82.9%(42)-----75.4%(43)-----62.0%(39)-----97.7%(87)------89.6%(131)\
> mRMR_BUR_CLF----96.7%(89)-------84.3%(37)-----75.2%(48)-----62.2%(47)-----98.6%(61)------89.4%(127)\
> ------------------------------------------------------------------------------------------------------------------------------------------\
> JMI----------------------96.8%(89)-------73.3%(17)-----74.1%(41)-----61.2%(31)------96.7%(86)------89.9%(134)\
> JMI_BUR---------------96.8%(89)-------73.9%(15)-----74.2%(37)------62.0%(35)-----96.8%(86)------89.4%(144)\
> JMI_BUR_CLF---------97.2%(93)-------73.6%(19)-----75.3%(49)------61.8%(37)-----98.1%(75)------90.0%(119)\
> -------------------------------------------------------------------------------------------------------------------------------------------\
> JMIM-------------------96.7%(98)--------76.8%(19)-----73.3%(49)-----62.2%(41)-----96.8%(93)------88.5%(138)\
> JMIM_BUR------------96.7%(98)--------77.5%(23)-----73.3%(38)-----62.3%(44)-----97.2%(94)------88.6%(144)\
> JMIM_BUR_CLF------97.3%(95)--------79.6%(27)-----73.6%(37)-----62.4%(43)-----98.4%(71)------89.2%(133)\
> -------------------------------------------------------------------------------------------------------------------------------------------\
> GSA---------------------96.4%(95)-------73.7%(35)------73.8%(36)-----63.5%(39)-----96.8%(67)------89.3%(138)\
> GSA_BUR--------------96.4%(95)-------74.2%(27)------74.1%(31)-----63.2%(33)-----97.1%(54)------89.7%(131)\
> GSA_BUR_CLF--------96.8%(96)-------76.3%(32)------73.9%(46)-----63.7%(37)-----98.6%(68)------90.0%(137)\
> ---------------------------------------------------------------------------------------------------------------------------------------------\
> RFECV------------------96.3%(103)------80.7%(35)------73.0%(30)-----61.9%(31)-----97.4%(58)-----88.5%(129)\
> =======================================================================================

---

### Author Response · Authors · 2020-11-21
**General Response Regarding the Differences from Previous Works**

Thanks to all reviewers for their critical reviews and insightful feedback. We apologize for the overlap with previous results and agree we could have done much better (even though we did cite all relevant previous works). In the revised paper, we will remove the overlapping content and highlight the differences with existing work. In this general response, we would like to highlight that our work is different from [Liu, 2018] and [Liu, 2020] and does move our understanding of the problem area forward.

**1. On the Major Differences with [Liu, 2020]:**

[Liu, 2020] mainly focuses on understanding the objective of MIBFS. For example, [Liu, 2020] suggested why minimal feature subset is preferred and demonstrated the drawback of irrelevance. [Liu, 2020] is a preliminary work and the very first step towards the MRwMR-BUR criterion. Our current work, which uses the same definitions as [Liu, 2020], is a follow-up work which addresses several important limitations of the analysis and approach in [Liu, 2020] and makes the following contributions.

__(i) The important finding that existing MRwMR based algorithms are underperforming:__ [Liu, 2020] does NOT investigate how existing MRwMR based algorithms select features with UR. In our work (see Fig. 2 and Table. 3 in Appendix), we calculate the redundancy rate for various MRwMR based algorithms and demonstrate that all of them are underperforming due to their high redundancy rates. We note that the results in Fig. 2 and Table 3, 4, 5 are not in [Liu, 2020].

__(ii) The motivating finding that prioritizing UR helps to achieve the objective of MIBFS:__ [Liu, 2020] does NOT study why UR should be prioritized. In our work (Section 6), we demonstrate that prioritizing features with UR can significantly reduce the redundancy rate. This provides a solid motivation for the MRwMR-BUR criterion. We will expand on this point and provide more evidence in the revised paper.

__(iii) The New CLF Variant:__ [Liu, 2020] estimates UR using the KSG estimator. The estimation of UR in such a  manner only provides marginal improvement in some linear classifiers (e.g., linear SVM). For example, in the table below, the improvement of MRwMR-BUR over MRwMR using linear SVM is marginal. We posit this is because MI captures the non-linear relationship which may not be favored by the linear classifier. This is the major drawback of the MRwMR-BUR in [Liu, 2020]. Our work addresses the drawback by proposing a classifier based approach to estimate UR. Our results demonstrate that this further improves the performance of the MRwMR-BUR criterion over [Liu, 2020].

__(iv) Comparison to another wrapper method (New Experiment):__ We note that the CLF variant changes MIBFS to a semi-wrapper method. Therefore, we conduct new experiments to compare its performance to another wrapper method (i.e., recursive feature elimination with cross validation (RFECV)).

**2. On the Differences with [Liu, 2018]:**

[Liu, 2018] proposed a MIBFS algorithm called SURI, which incorporates UR. Our work is NOT proposing a new MIBFS algorithm; rather we are motivating MRwMR-BUR as a general MIBFS criterion (we note that this includes SURI which is referred to as GSA-BUR). Specifically, we provide additional analysis on why features with UR should be prioritized. More importantly, we propose a new method to estimate UR which promotes the use of MRwMR-BUR and helps to improve the performance.

---

### Decision · Program_Chairs · 2021-01-07
**Final Decision**

**Decision:**

Reject

**Comment:**

During the discussion among reviewers, we have shared the concern that this work has a significant overlap with [Liu et al. 2018] and [Liu & Motani 2020]. Although the authors tried to address this concern by the author response, I also think that the difference is not enough. In particular, the reviewers pointed out that Figure 1, Table 1, and Figure 3 are exactly the same with those in [Liu, 2020], and Proposition 2 in [Liu & Motani 2020] is Proposition 1 in this paper. Since these overlaps are not acceptable, I will reject the paper.